# Effect of Electrode Spacing on the Performance of a Membrane-Less Microbial Fuel Cell with Magnetite as an Additive

**DOI:** 10.3390/molecules28062853

**Published:** 2023-03-22

**Authors:** Nhlanganiso Ivan Madondo, Sudesh Rathilal, Babatunde Femi Bakare, Emmanuel Kweinor Tetteh

**Affiliations:** 1Green Engineering Research Group, Department of Chemical Engineering, Faculty of Engineering and the Built Environment, Durban University of Technology, Steve Biko Campus, S4 Level 1, Durban 4000, South Africa; rathilals@dut.ac.za (S.R.); emmanuelk@dut.ac.za (E.K.T.); 2Environmental Pollution and Remediation Research Group, Department of Chemical Engineering, Faculty of Engineering, Mangosuthu University of Technology, P.O. Box 12363, Durban 4026, South Africa; bfemi@mut.ac.za

**Keywords:** electrode spacing, microbial fuel cell, bioelectrochemical system, magnetite nanoparticles, anaerobic digestion

## Abstract

A microbial fuel cell (MFC) is a bioelectrochemical system that can be employed for the generation of electrical energy under microbial activity during wastewater treatment practices. The optimization of electrode spacing is perhaps key to enhancing the performance of an MFC. In this study, electrode spacing was evaluated to determine its effect on the performance of MFCs. The experimental work was conducted utilizing batch digesters with electrode spacings of 2.0 cm, 4.0 cm, 6.0 cm, and 8.0 cm. The results demonstrate that the performance of the MFC improved when the electrode spacing increased from 2.0 to 6.0 cm. However, the efficiency decreased after 6.0 cm. The digester with an electrode spacing of 6.0 cm enhanced the efficiency of the MFC, which led to smaller internal resistance and greater biogas production of 662.4 mL/g VS_fed_. The electrochemical efficiency analysis demonstrated higher coulombic efficiency (68.7%) and electrical conductivity (177.9 µS/cm) for the 6.0 cm, which was evident from the enrichment of electrochemically active microorganisms. With regards to toxic contaminant removal, the same digester also performed well, revealing removals of over 83% for chemical oxygen demand (COD), total solids (TS), total suspended solids (TSS), and volatile solids (VS). Therefore, these results indicate that electrode spacing is a factor affecting the performance of an MFC, with an electrode spacing of 6.0 cm revealing the greatest potential to maximize biogas generation and the degradability of wastewater biochemical matter.

## 1. Introduction

Worldwide, the ever-increasing standard of living and population are fundamental causes of the rise in energy demand [1]. The increase in the consumption of energy leads to the deterioration of natural resources. To preserve natural resources, many efforts have to be made to reduce the negative influences of energy generation [2,3,4]. Over the past few years, special consideration has been made to other sources of energy that are eco-friendly and abundant [5]. Unfortunately, the generation of energy from several sources of renewable energy completely relies on environmental conditions, making these sources almost inappropriate for addressing peak demands. Consequently, continuous energy provision cannot be made certain only by these sources, and new renewable energy techniques are needed [6,7,8]. 

The bioelectrochemical system (BES) is a promising renewable energy technique that treats wastewater while at the same time generating energy, wherein the microbes of the anode oxidize biochemical matter and produce a flow of electricity into the system [9]. The use of BES techniques has grown from the generation of electrical energy to the generation of hydrogen, the synthesis of organic and inorganic compounds, and the promotion of desalination or remediation of contaminants [10]. The various types of BES techniques have one thing in common; the anodic semi-cell mechanism depends on exoelectrogenic microorganisms [11]. Optimizing the capability of exoelectrogenic microorganisms in the anode is, therefore, a central feature for enhancing the total performance of a BES [12]. 

Microbial fuel cells (MFC) and microbial electrolysis cells (MEC) have emerged as promising BES technologies [13]. A typical BES involves an anodic section, where microorganisms oxidize the biochemical substrate and transfer electrons externally, and a cathodic section, where protons and electrons catalytically react during the reduction in molecular oxygen to produce H_2_O (MFC) or generate hydrogen (MEC) [14]. The main difference between the two BESs is that the MEC needs an electrical current flowing externally to make certain that the cathodic mechanism is highly favorable. With regards to an MEC, the aim is to produce methane or hydrogen, which in this specific instance is formed in the cathode section. The best-performing BES is the MEC; our previous study revealed a higher maximum current density of 23.33 mA/m^2^ on the MEC compared to the 7.5 mA/m^2^ on the MFC [15]. Despite its better performance, the MEC has received limited use due to high energy usage. 

In view of this, the MFC has fundamental benefits over methods conventionally used for the treatment of wastewater: (a) the MFC can be used for the conversion of biochemical matter into electrical energy, preventing energy conversion losses obtained in other methods; (b) the MFC functions over a wide span of pH values and temperatures (such as thermophilic and mesophilic temperatures); (c) the MFC can be employed with numerous biochemical matter; and (d) the MEC has a somewhat lower generation rate of sludge [16,17]. MFC systems are emerging as multipurpose renewable energy systems. This is mainly due to the multidimensional usages of this environmentally friendly system. The MFC is dependent on electrically active microorganisms, otherwise known as exoelectrogens, for the generation of electricity and the treatment of wastewater [18,19]. 

The main disadvantage of the MFC system is the low density of power production obtained by this system [20]. Even though MFCs have gained attention since the commencement of this century, the ability of these systems to generate power has not improved significantly within this timeframe. Scale-up by enlarging the size of the electrodes with the aim of improving power generation has been unsuccessful since wrong assumptions were made that a linear function could express the correlation between the power generated by an MFC and the size of the digester. Instead, power generation can be enhanced by optimizing process variables as opposed to scaling up. The power output of an MFC could be enhanced by optimizing process variables, for example, temperature, flow rate, pH, COD, or configuration options, for instance, electrode spacing [6]. Electrode spacing is perhaps one of the most important variables of a BES. This is because electrode spacing affects the current density and overall resistance in the system [21]. Considering that one of the main issues of an MFC is optimizing the power generation, the study of the influence of electrode spacing thereon is very significant for researchers [22,23].

Carbonaceous conductive materials, such as activated carbon, biochar, carbon cloth, magnetite, and granular activated carbon, may function to enhance the direct species electron transfer (DIET) process in anaerobic digestion as they help in the acceleration of volatile fatty acid accumulation and offer substrates to methanogens [24]. The employment of conductive additives that contain iron in anaerobic digestion may be beneficial for improving the interspecies electron transfer between hydrogenotrophic methanogens and volatile fatty acid-oxidizing microorganisms. In anaerobic digestion, the majority of enzymes are metalloenzymes; thus, they require the existence of metals, such as co-enzymes. Actually, almost all metalloenzymes that are in the path of biogas formation have numerous agglomerates containing iron; iron is certainly vital for methane production and cytochromes. Since iron plays an important role in the transportation of electrons, both hydrogen and methane generation rates improve, and the growth of bacteria is increased by stimulating the actions of enzymes [25]. Therefore, magnetite, an iron oxide, is the most customary material compared to any other carbonaceous conductive materials because of its higher biocompatibility and lower toxicity [26]. In fact, several reports have suggested magnetite nanoparticles to be a promising anaerobic additive in bioelectrochemical systems such as MFCs, because they promote interspecies transfer between microorganisms and Archaea, with the improved diversity of microorganisms in the digester [27,28,29]. Our previous work [15] examined the influence of the addition of 1 g of magnetite nanoparticles on a MEC. The outcome of the study revealed that the supplementation of magnetite-nanoparticles enhanced biogas generation (15 mL) and the percentage of methane (84%) by about 3.1 and 1.1 times the 4.8 mL and 79.1% of the MEC, respectively. Moreover, the use of magnetite-nanoparticles improved the flow of the current; highly active microbes (such as exoelectrogens) flowed to the anodic electrode and then to the cathodic electrode, thus enhancing the flow of electrons by 7.3%. Table 1 shows recent studies on MFCs. The studies have focused on MFCs with a TiO_2_, honeycomb-type flow straightener, or metal-metal oxides. However, there have been very few studies on the use of MFCs and magnetite nanoparticles.

Electrode spacing has a significant influence on the efficiency of an MFC [33]. As the electrode spacing increases, the total distance that the protons generate by electrochemically active microbes that need to move from the anode electrode to the cathode electrode increases, which ultimately increases the mechanism system’s resistance [34,35]. On the other hand, very small electrode spacing will allow some of the oxygen present in the air to enter the anodic electrode and undergo a reaction with the electrochemically active microbes. As a result, several electrons are consumed, which leads to higher internal resistance and a reduced output current [36]. Based on this, optimum electrode spacing is needed to increase the performance of an MFC in anaerobic digestion.

Therefore, this study explored the effect of electrode spacing on the performance of a membrane-less MFC with magnetite as an additive. Special attention was paid to biogas accumulation, electrochemical characteristics, the pH of the system, and the removal of toxic contaminants. 

## 2. Results and Discussion

### 2.1. Biogas Generation

In anaerobic digestion, the graph of biogas accumulation is very important for observing the microbial growth rate of the system [37]. The relationship between biogas accumulation and time was investigated using 2 cm, 4 cm, 6 cm, 8 cm, and control, and the results are graphically represented in Figure 1. As evident from the graph, biogas accumulation was extremely slow at the early stages of the digestion period for all digesters. Before day 5, the MFC digester with the smallest electrode spacing of 2 cm performed better than the other digesters as it revealed the highest biogas accumulation of 160 mL/g VS_fed_. After passing the hydrolysis stage, the digesters experienced a high rate of biogas accumulation. On day 6, the biogas accumulation of the MFC digester with an electrode spacing of 6 cm coincided with that of the MFC digesters with an electrode spacing of 2 and 4 cm. After day 6, a higher upward trend in the biogas accumulation of the digester with 6 cm electrode spacing was observed. After experiencing a high increase in the methanogen growth rate, the biogas accumulation of all digesters decreased. Then, all digesters stabilized as they approached the asymptotic stage. In the end, the MFC digester with an electrode spacing of 6 cm generated the highest biogas generation of 662.4 mL/g VS_fed_, which was approximately 5.6 times more than the 117.7 mL/g VS_fed_ of the control. 

The graph of the daily biogas production that is displayed in Figure 2 gives more details on the growth rate stages. The digester with an electrode spacing of 8 cm had the longest lag phase of 5 days. This means that it took longer for the microorganisms to adapt to the electrodes of this digester. The distance between the electrodes of this digester was too large in such a way that it took longer for the protons and ions to travel from the anode to the cathode, and, hence, a lesser adaptability and biogas generation was observed. Once the microorganisms adapted to the system, the 8 cm MFC digester increased its biogas generation (i.e., as it approached the exponential stage) until, in the end, it generated a maximum biogas production of 82.0 mL/(g VS_fed_.day) on day 11, which was better than the 4 cm and 2 cm MFC digesters. On the other hand, the 6 cm MFC digester generated the highest biogas production of 89.2 mL/(g VS_fed_.day) on day 7. All digesters then approached the dead phase, where the daily biogas production decreased exponentially. Although the 2 cm MFC digester had the lowest lag phase as a result of low resistance, the high biogas accumulation of 53.0 mL/(g VS_fed_.day) on day 1 possibly resulted in volatile fatty acid (VFAs) accumulation, which made this digester reach the dead phase faster than the other digesters.

### 2.2. Current, Resistance and Voltage

As mentioned in the previous section, the graphs of biogas accumulation and daily biogas production over time are very useful in exploring the growth rate of bacteria. However, it is also important to examine the flow of electrons externally. This can be achieved by examining the parameters affecting Ohm’s law, such as the voltage generated and resistance. The DC voltage of the MFCs was recorded against time for a period of 30 days to follow the formation of biofilm on the surfaces of the anode electrodes (Figure 3). The voltage generated by the MFC with an electrode spacing of 2 cm rose quickly on the first day of experimentation, achieving a voltage of 0.135 V. The DC voltage of the digester with the widest electrode spacing of 8 cm gradually rose after 3 days and achieved the same voltage of 0.135 V after 6 days. These outcomes suggest that the biofilm grew better on smaller electrode spacing than on wider electrode spacing. The start-up period of the MFC digester, with an electrode spacing of 2 cm, was most likely improved by the lower resistance in this digester [38]. Certainly, Simeon et al. [39] found that larger electrode spacing delayed the attachment of biofilm, which ultimately made the MFC digester take a longer period to stabilize and optimize performance. Thus, it is likely that the longer start-up time for the 8 cm MFC digester is attributable to the higher resistance on this MFC digester [38]. However, this conclusion has to be verified in the resistance and electrochemical efficiency sections that will be discussed later in this paper.

The maximum voltage of 0.36 V was generated by the 6 cm MFC digester, which was about 1.6 times the 0.23 V of the least performing MFC (4.0 cm). Wu et al. [36] studied the influence of the spacing of electrodes on the generation of power using an MFC. As is the case with our study, these investigators also discovered that when the electrode spacing was set to 15, 30, and 60 cm, the highest voltage generation of 0.4 V was obtained at 6 cm. 

The total ohmic resistance of a microbial fuel cell is ideally proportional to the spacing between the anode and cathode electrodes [40]. The same conclusion was found in our investigation, as evident from Figure 4. The graph displays the same direct relationship between electrode spacing and resistance before day 10. After day 13, an increase in the overall resistance was seen on digesters with an electrode spacing of 2 cm and 4 cm. The rise in the overall resistance found on these digesters was mostly a result of the higher increase in resistance on charge transfer. This can be a result of fouling on the electrode, which is caused by a higher concentration of the substrate at the cathode electrode or by a high increase in the thickness of the biofilm at the anode section [41]. Of these two digesters, the digester with the smallest electrode spacing of 2 cm had the highest resistance at the end of the digestion period.

As was the case with voltage, the resistance of the MFC with 6 cm electrode spacing also stabilized from day 5 to day 15. After day 13, this digester revealed the lowest resistance. Therefore, it is the higher stability and the lower resistance (after day 13) that made the digester with an electrode spacing of 6 cm generate the highest voltage of 0.36 V and produce the highest biogas accumulation of 662.4 mL/g VS_fed_. 

### 2.3. Electrochemical Efficiencies

The electrochemical characteristics of the MFC digesters were represented by power density, electrical conductivity, and coulombic efficiency. One of the most important factors of an MFC is power density. In an MFC, in the same way as with other power generators, the main aim is to optimize the output power [42]. The power density is affected by electrode spacing [43]. From Figure 5, all MFC digesters first showed an increase in power density, reached a maximum power density, and lastly showed a decrease in power density. In the early stages of the digestion process, the power density significantly improved as the spacing of the electrode narrowed. This outcome revealed the significance of electrode spacing to the power density of an MFC, which was possible as a result of a reduction in ohmic resistances when the distance between the electrodes was reduced [44]. Smaller distances resulted in a small resistance in the early stages of the digestion process and a more notable enhancement in the power density [45]. However, after day 5, the digester with an electrode spacing of 6 cm surpassed that of the smaller distance.

The increasing order of the maximum power density indicated the following: 2 cm: 26.2 mW/m^2^ > 6 cm: 24.7 mW/m^2^ > 8 cm: 16.9 mW/m^2^ > 4 cm: 14.7 mW/m^2^. Therefore, the digester with an electrode spacing of 2 cm revealed the greatest performance as it showed the highest maximum power density of 26.2 mW/m^2^. Even though the highest maximum power density was obtained at a 2 cm MFC digester as a result of the smaller resistance and easy accessibility of the substrate, the influence of oxygen diffusion was very high, which led to a sudden fall in its power density [41]. In the end, this digester did not stabilize at all. 

Instead, the MFC digesters with an electrode spacing of 4 cm, 6 cm, and 8 cm stabilized. The digester with an electrode spacing of 6 cm stabilized at the highest power density of 24.7 mW/m^2^ (from days 20 to 30). 

Figure 6 shows the average power density and accumulation concerning resistance. From the result shown in Figure 6, it can be elucidated that a decrease in resistance increased both the average power density and biogas accumulation. On the other hand, power density was lesser affected by high resistances (117.23–120.89 Ω); it was greatly affected by smaller resistance. The digester with the lowest average resistance of 113.07 Ω generated the highest average power density of 11.56 mW/m^2^ and the greatest biogas accumulation of 662.4 mL/g VS_fed_. Furthermore, it is evident from the figure that for very high resistances (117.23–120.89 Ω), biogas production was highly enhanced by even a very small increase in power generation. However, for a very low resistance of 113.07 Ω (6 cm), biogas accumulation was lesser affected by power density.

Coulombic efficiency is the efficiency with which electrons are transported in a process to bring the electrochemical reaction to completion. This is an important measure of the performance of MFCs as it calculates the total number of coulombs retrieved as an electrical current [46]. Coulombic efficiency and electrical conductivity as a function of electrode spacing are depicted in Figure 7. For electrode spacing below 6 cm, both electrical conductivity and coulombic efficiency improved with a rise in electrode spacing; on the other hand, beyond 6 cm, electrode spacing hindered the performance of the MFC digesters. As was the case above, the MFC digester with an electrode spacing of 6 cm performed better, as revealed by the higher electrical conductivity and coulombic efficiency. The coulombic efficiency depends on the process’s resistance, and higher resistance leads to a decrease in the coulombic efficiency since it is not easy to recover electrons from a system with very high resistance [47]. Thus, this implied that the 6 cm MFC digester had the lowest internal resistance; the high coulombic efficiency indicates that this MFC retrieved the greatest number of electrons. 

Figure 8 portrays the influence of electrical conductivity on both coulombic efficiency and maximum power density. It can be depicted from the figure that increasing electrical conductivity resulted in an increase in both the average power density and coulombic efficiency. Therefore, it is the great electrical conductivity of the digester with an electrode spacing of 6 cm that caused this digester to produce the greatest coulombic efficiency of 68.7% and an average power density of 11.56 mW/m^2^. The greater electrical conductivity of 177.9 µS/cm definitely improved the performance of this MFC digester since it decreased ohmic losses (i.e., the opposition or resistance that protons and ions face as they move in the system) in the MFC digester. Thus, the higher protons and ions present in the solution enhanced the flow of the current in the external circuit [15] and, subsequently, the average power density.

### 2.4. The pH of the System

The pH of the system was used as a stability indicator, i.e., it was used to observe the health of the system. With pH as a function of time (Figure 9), an increase in electrode spacing in the early stages of the digestion process (i.e., before day 5) was accompanied by a decrease in pH. According to our previous study [15], higher pH enhances the performance of the bioelectrochemical system. The digester with 2 cm spacing performed better in the early stages of the digestion process as it revealed a higher pH. It was perhaps due to the small spacing that the movement of protons from the anode electrode to the cathode electrode was facilitated, thus enhancing the performance of this digester before day 5. However, the high movement of protons, together with the high growth rate on this digester (as was evident in Figure 1), possibly resulted in higher biofilm growth on the electrode, leading to a decrease in pH after day 5.

The average pH values for electrode spacings of 2, 4, 8, and 6 cm were 7.35, 7.48, 7.57, and 7.69, whereas the average power densities for biogas accumulation were 5.51, 5.88, 5.99, and 11.56 mW/m^2^, respectively. Thus, the higher the pH, the higher the power density; the digester with the highest pH of 7.69 (i.e., electrode spacing of 6 cm) generated the highest average power density of 11.56 mW/m^2^. 

### 2.5. Removal of Toxic Contaminants 

The treatability efficiency of anaerobic digestion was expressed in terms of the water quality parameters, viz. color, COD, TSS, VS, and TS removals, as depicted in Figure 10. All bioelectrochemical systems performed better than the traditional anaerobic digestion process (i.e., control), revealing a contaminants removal of over 69%. This proved that the synergistic use of MFC and magnetite-nanoparticles enhanced the treatability of the traditional anaerobic process [48]. The MFC digester with 6.0 cm electrode spacing performed better, with over 83% for COD (87.7%), TS (86.1%), TSS (83.2%), and VS (86.2%). This could be attributed to the higher stability of this digester, which eventually led to improved electron recovery efficiencies, as indicated by higher electrical conductivity (177.9 µS/cm) and maximum voltage (0.36 V), and this resulted in higher contaminant removals.

By contrast, color removal (82.1%) was better on the digester with an electrode spacing of 2 cm. In fact, as evidenced in Table 2, the color removals of 17, 77, 78, 81.1, and 82.1% were achieved for magnetic field strengths of 0.3, 4.8, 5.45, 5.89, and 6.5 mT. Therefore, an increase in the magnetic field strength resulted in an increase in color removals. Our previous findings [49] revealed that exposure to higher magnetic field strength improved the removal of color in wastewater. The same conclusion was found by other researchers who also reported that the quality of treated water in terms of color would be greater if the magnetic field strength used was also greater [50,51]. Therefore, it is the extremely high magnetic field strength of 6,50 mT on the digester with an electrode spacing of 2 cm that made it remove more color than the other digesters. 

## 3. Materials and Methods

### 3.1. Equipment Set-Up and Operation

The experimental work was performed in the Water Membrane Laboratory at the Durban University of Technology (DUT) to determine the impact of electrode spacing on the anaerobic digestion of sewage sludge. The electrode spacing was expressed as the distance between the anodic electrode and the cathodic electrode plates of an MFC. Batch digesters with a total volume of 1 L were operated at a mesophilic temperature of 32.2 °C [52] for a hydraulic retention time of 30 days (Figure 11). The working volume of each digester was 0.8 L, which consisted of 0.3 L of sewage sludge, 0.5 L of waste-activated sludge, and 0.53 g of magnetite nanoparticles [52]. The electrode spacing of the MFCs was set at different values: 1.0, 2.0, 4.0, and 6.0 cm. Two rectangular-shaped electrodes with a length of 12 cm and a width of 1 cm were inserted upright inside each digester. The anode section consisted of a zinc electrode, whereas the cathode section consisted of a copper electrode. In order to close the circuits of the MFCs, the anode and cathode electrodes of each of the digesters were connected to a 100 Ω external resistor by means of wires. The top cap of each digester consisted of four ports for feeding, an anode electrode, a cathode electrode, and a biogas line. The generated gas, i.e., biogas, was transferred to a biogas collector via a 0.6 cm tubing connector.

### 3.2. Sample Analyses and Substrates 

A water-displacement system was used to measure the biogas that was generated daily. Chemical oxygen demand (COD), total solids (TS), total suspended solids (TSS), volatile solids (VS), and color are the water quality quantities that were investigated. Color, TSS, and COD were measured on the first day and last day using a Hach DR 3900 colorimeter (Hach, Loveland, CO, USA). Wastewater was characterized via the standard methods proposed by APHA [53]. A FLUKE 177 RMS digital multimeter (RMS, Everett, WA, USA) was used to acquire the resistance, total current, and voltage generated, which were measured daily. A Hanna H198129 conductivity meter was used to measure the pH, electrical conductivity, and total dissolved solids (TDS) once every 5 days. Magnetic field strength measurements were acquired using a digital Telsameter. Table 3 shows the physio-chemical properties of the effluent. 

The current density, power density, coulombic efficiency, and percentage of toxic contaminants removed were calculated as previously explained [54].

Waste-activated sludge was used as a substrate, while sewage sludge was used as an inoculum, and both were taken from treatment works based locally in Durban, South Africa. Containers with a capacity of 20 L were used for sampling the wastewater.

### 3.3. Magnetite-Nanoparticles Synthesis 

The magnetite nanoparticles that were employed in this investigation were obtained from the magnetite nanoparticles produced by Amo-Duodu et al. [55]. The work included the analysis of magnetite-nanoparticles morphological and physiochemical features, which were obtained using scanning electron microscopy/energy-dispersive X-ray (SEM/EDX), X-ray diffraction (XRD), and Fourier-transform infrared spectroscopy (FTIR). A co-precipitation technique was used to synthesize magnetite-nanoparticles that involved adding chemical reagents, viz. nickel (II) nitrate hexahydrate, ferrous sulfate heptahydrate, sodium hydroxide, ferric chloride hexahydrate, and oleic acid. Ferric chloride hexahydrate was purchased from United Scientific SA cc, South Africa. Both nickel (II) nitrate hexahydrate and oleic acid were obtained from Sigma–Aldrich, South Africa, whereas both sodium hydroxide and ferrous sulfate heptahydrate were attained from Labcare Supplies (PTY) LTD. The structure of magnetite-nanoparticles was verified to be a face-centered cubic shape using XRD, whereas the crystal size was found to be 5.179 nm. 

## 4. Conclusions

An extensive study was conducted to determine the influence of electrode spacing on anaerobic digestion performance. From the results attained within this investigation, the following conclusions can be taken. Electrode spacing affected the parameters of interest, namely biogas accumulation, electrochemical characteristics, the pH of the system, and removal of toxic contaminants. The digester with the lowest electrode spacing of 2 cm performed better in the early stages of the digestion process (before day 5), revealing a high biogas accumulation of 160 mL/g VS_fed_. The high generation of this digester was possible by forming higher VFAs in the system, as indicated by the lower pH, which ultimately made the digester fail after day 5. Nonetheless, the highest biogas production of 662.4 mL/g VS_fed_ was found when the electrode spacing was set to 6.0 cm. The electrochemical analysis also showed a higher coulombic efficiency of 68.7% and electrical conductivity of 177.9 µS/cm. The digester with 6.0 cm electrode spacing proved most efficient as it generated the highest voltage power of 0.36 V, which was about 1.6 times the 0.23 V of the least performing MFC (4.0 cm). In terms of toxic contaminants removed, the MFC digester with 6.0 cm electrode spacing also performed the best, with removals of 87.7%, 86.1%, 83.2%, and 86.2% for COD, TS, TSS, and VS, respectively. Overall, the MFC with an electrode spacing of 6.0 cm showed a better possibility for use in bioelectrochemical biogas production and treatment of wastewater.

## Figures and Tables

**Figure 1 molecules-28-02853-f001:**
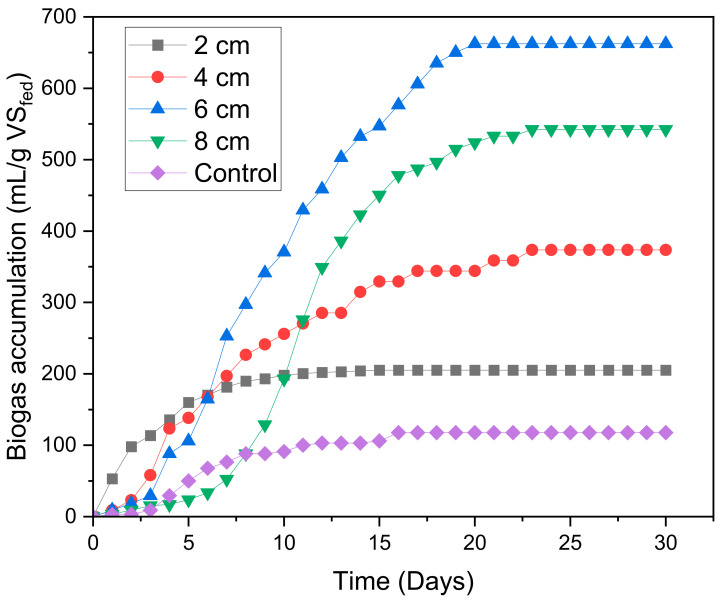
Biogas accumulation as a function of time.

**Figure 2 molecules-28-02853-f002:**
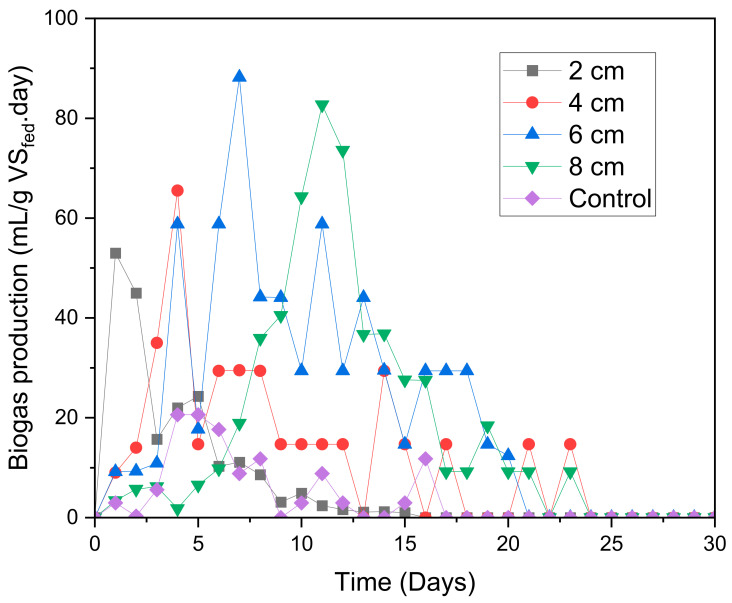
Daily biogas production.

**Figure 3 molecules-28-02853-f003:**
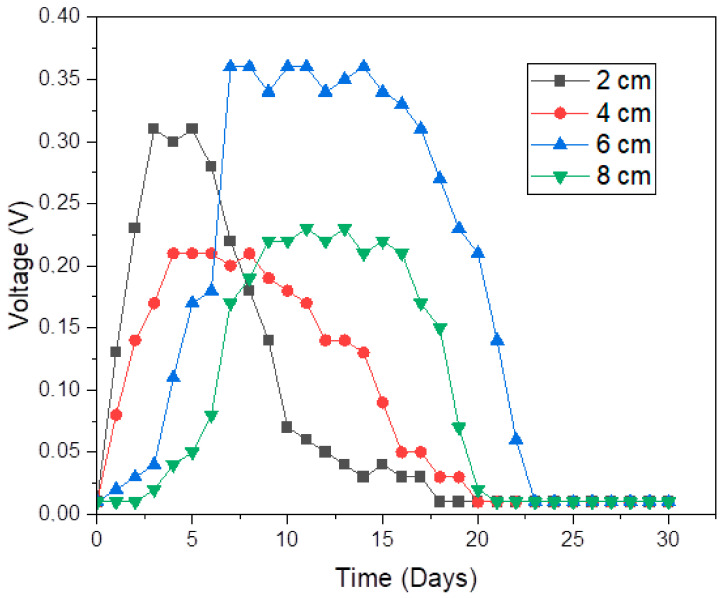
Voltage generation over time.

**Figure 4 molecules-28-02853-f004:**
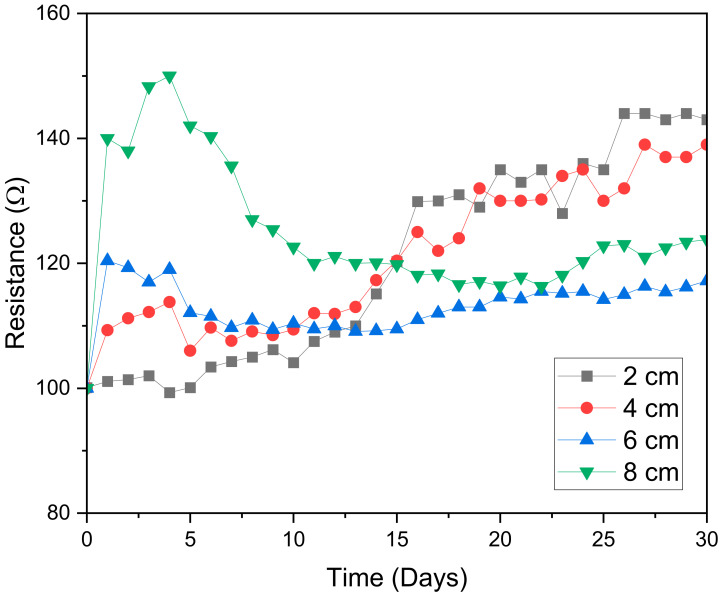
Variation in total resistance with time.

**Figure 5 molecules-28-02853-f005:**
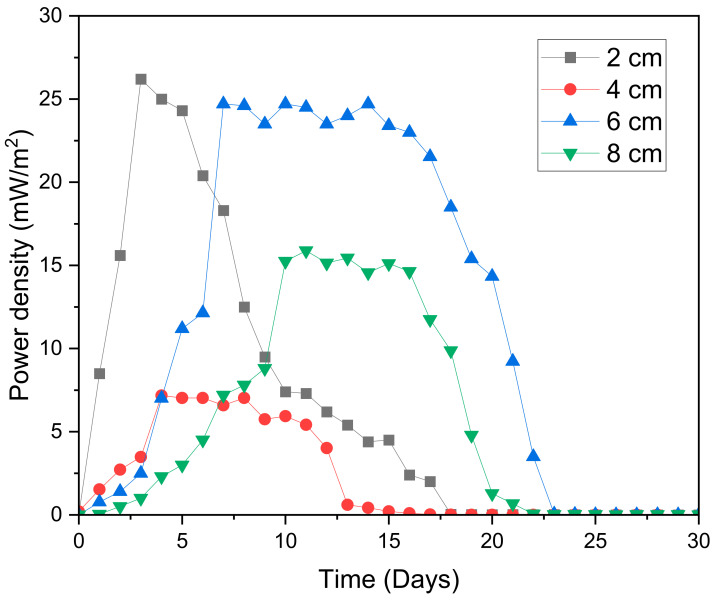
Power density as a function of time.

**Figure 6 molecules-28-02853-f006:**
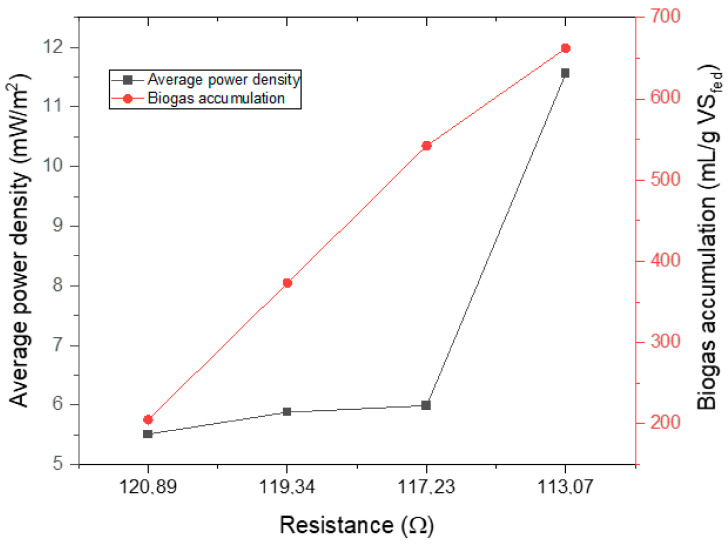
Average power density and biogas accumulation with respect to resistance.

**Figure 7 molecules-28-02853-f007:**
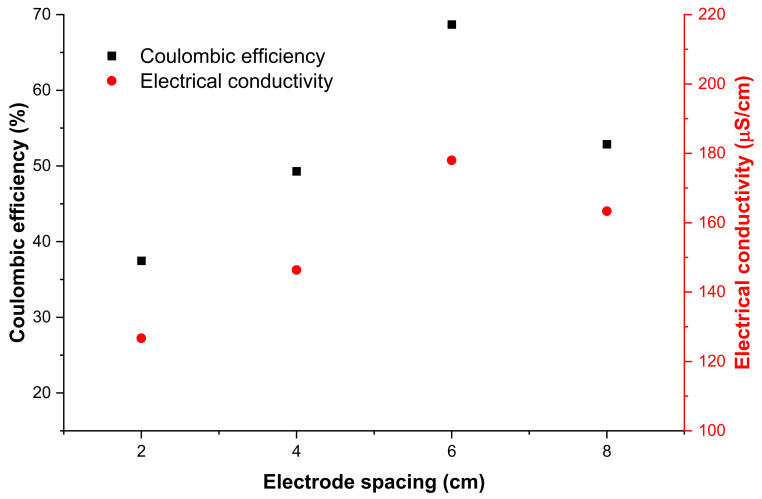
Coulombic efficiency and electrical conductivity as a function of electrode spacing.

**Figure 8 molecules-28-02853-f008:**
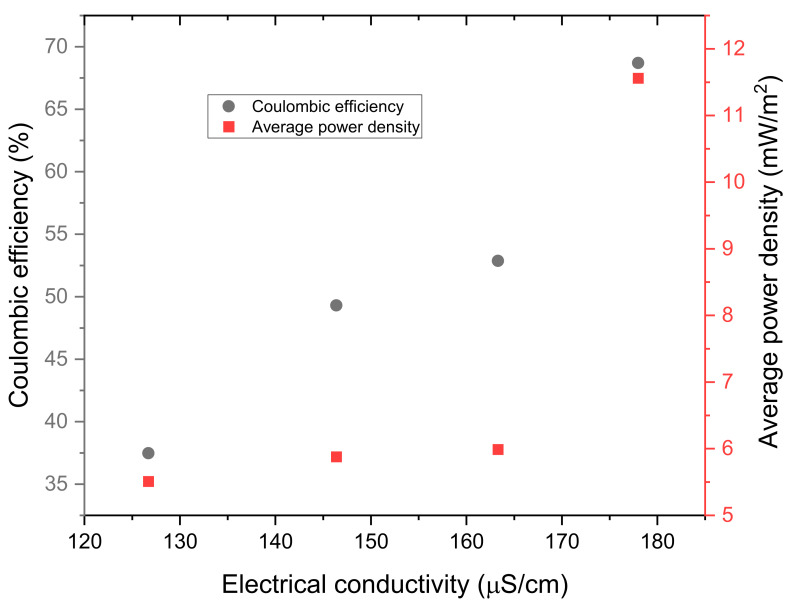
Coulombic efficiency and maximum power density as a function of electrical conductivity.

**Figure 9 molecules-28-02853-f009:**
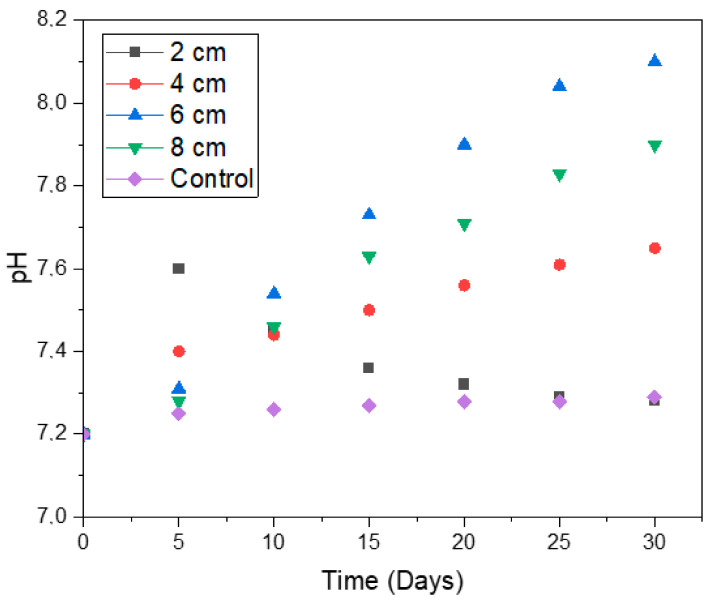
The pH of the digesters with respect to time.

**Figure 10 molecules-28-02853-f010:**
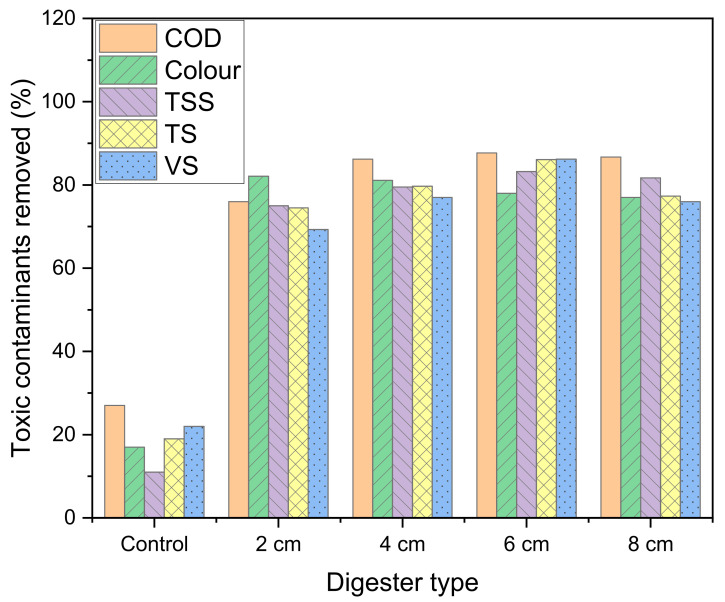
Influence of electrode spacing on toxic contaminants removal.

**Figure 11 molecules-28-02853-f011:**
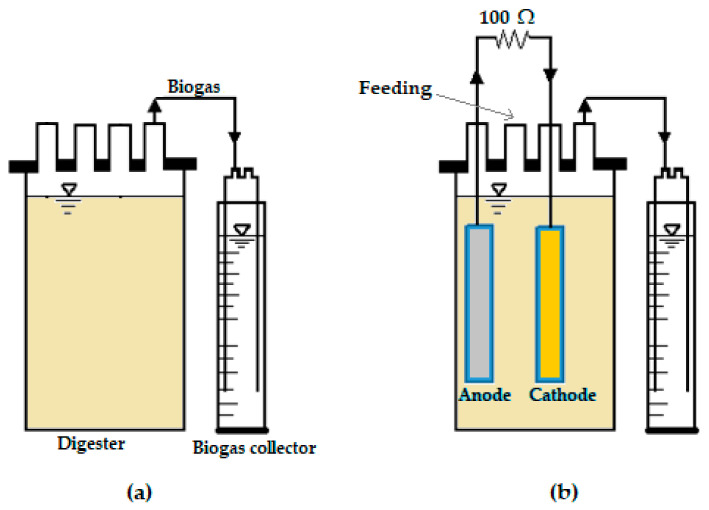
Schematic representation of the equipment set-up for: (**a**) Control; and (**b**) MFC.

**Table 1 molecules-28-02853-t001:** Recent studies on MFCs.

Type of Bioelectrochemical System	Additive/Device Used for Improved Performance	Reference
MFC	TiO_2_	Khan et al. [30]
MFC	Honeycomb type flow straightener	Wang et al. [31]
MFC	Metal-metal oxides	Ali et al. [32]

**Table 2 molecules-28-02853-t002:** Influence of magnetic field strength on color removal.

Digester Type	Magnetic Field Strength (mT)	Color Removal (%)
control	0.30	17.0
8 cm	4.80	77.0
6 cm	5.45	78.0
4 cm	5.89	81.1
2 cm	6.50	82.1

**Table 3 molecules-28-02853-t003:** Physio-chemical properties of the influent.

Parameter	Unit	Amount
pH	-	7.20 ± 0.32
Color	Pt.Co.	254.33 ± 6.30
COD	mg/L	2451.23 ± 200.45
TS	mg/L	53.34 ± 5.34
VS	mg/L	45.55 ± 1.12
TSS	mg/L	39.35 ± 1.06

## Data Availability

Not applicable.

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
