# Peer review of "Effect of Electrode Spacing on the Performance of a Membrane-Less Microbial Fuel Cell with Magnetite as an Additive"

_molecules, 2023, doi:10.3390/molecules28062853_

Round 1

Reviewer 1 Report

Major Revision

(molecules-2244724)

The submitted manuscript (molecules-2244724) “Effect of Electrode Spacing on the Performance of a Membrane- 2 less Microbial Fuel Cell with Magnetite as an Additive” This article focused on microbial fuel cell and it use in generation of electrical energy under microbial activity during wastewater treatment practices.

The author claimed that the electrochemical efficiency analysis demonstrated higher coulombic efficiency (68.7%) and electrical conductivity (177.9 µS/cm) for the 6.0 cm, which is evidence for the enrichment of electrochemically-active microorganisms. With regards to toxic contaminant removal, the same digester also performed well, revealing removals of over 83% for chemical oxygen demand (COD), total solids (TS), total suspended solids (TSS), and volatile solids (VS).

However, the article is well written with reasonable results and explanation but still needs to be revised with minor errors before accepted for publication.

 Reviewer comment;

1. Regarding originality and scientific progress of this work, in the introduction section, author needs to include more information and mention the advantages of magnetite over other materials used in MFC.

2. Brief explanation needs to disadvantage of the MFC system is the low-density of power production obtained by this system. How to overcome electrode spacing that affects the current density and overall resistance in the system?

3. Line no. 101 “[Error! Reference source not found.]. similarly in line 124, 152, 166, 190, 212, 233, 255, 256, 272, 282, 299, 307, 320, 332,

4. The manuscript lacks of innovation, the author must consider the given article to revise the introduction section and cite these references at appropriate places.

5. The author needs to add recent development of research in tabular form to compare their work for novelty in the introduction section.

doi.org/10.1038/s41598-018-19617-2; DOI: 10.1007/978-3-030-79899-4_3; doi.org/10.1002/er.5776;

6. 30 days of period will be enough for formation of biofilm? Or is it optimized?

7.  The role of pH is also important as a function of time for the digestion process. So how many days required to optimized pH values. I mean increase or decrease the pH value?

8. The conclusion must be precise as per the findings.

9. There are few spellings, punctuation, and formatting errors in this article and it is recommended that the entire article be carefully revised.

10. The author should make the necessary modifications, addition, citation and responses to the queries raised above before acceptance.

Author Response

Thank you so much for your comments. Kindly find attached responses to your comments.

Reviewer 2 Report

The authors present a study about the electrode spacing used in MFC to determine its effect on performance. The experimental work was conducted utilizing digesters with different electrode spacings  The results demonstrated that the performance of the MFC improved with an electrode spacing of 6.0 cm enhancing the efficiency of an MFC, furthermore, reached a smaller internal resistance and greater biogas production. The electrochemical efficiency analysis for the 6.0 cm demonstrated higher coulombic efficiency and electrical conductivity, which is evidence for the enrichment of electrochemically-active microorganisms. With regards to toxic contaminant removal, revealed removals for chemical oxygen demand, total solids, total suspended solids, and volatile solids. In my opinion, the paper is interesting; however, I have some comments and questions.

To facilitate the reading of the text it is necessary that the authors properly cite the references, apparently, your compiler lost the sequence of the references.

Figure 7 can improve the presentation, in particular the X axis, is not very clear.

What impact does the material of the anode and cathode have on the behavior of the digester?

The control biodigester must contain the electrodes in open mode, without measuring the energy generated, in order to know the effect of the biogas only with the presence of the electrode materials.

What relevant role do magnetic nanoparticles play in the performance of energy and biogas generation?

Author Response

(The authors gave the same response as above.)

Round 2

Reviewer 1 Report

The author well addressed all the queries raised by the reviewers. 

Reviewer 2 Report

The authors have improved the presentation of the brief.